# GAUSSIAN MRF COVARIANCE MODELING FOR EFFICIENT BLACK-BOX ADVERSARIAL ATTACKS

## ABSTRACT

We study the problem of generating adversarial examples in a black-box setting, where we only have access to a zeroth order oracle, providing us with loss function evaluations. We employ Markov Random Fields (MRF) to exploit the structure of input data to systematically model the covariance structure of the gradients. The MRF structure in addition to Bayesian inference for the gradients facilitates one-step attacks akin to Fast Gradient Sign Method (FGSM) albeit in the black-box setting. The resulting method uses fewer queries than the current state of the art to achieve comparable performance. In particular, in the regime of lower query budgets, we show that our method is particularly effective in terms of fewer average queries with high attack accuracy while employing one-step attacks.

## 1 INTRODUCTION

Most methods for adversarial attacks on deep learning models operate in the so-called *white-box* setting (Goodfellow et al., 2014), where the model being attacked, and its gradients, are assumed to be fully known. Recently, however there has also been considerable attention given to the *black-box* setting as well, where the model is unknown and can only be queried by a user, and which much better captures the "typical" state by which an attack can interact with a model (Chen et al., 2017; Tu et al., 2019; Ilyas et al., 2019; Moon et al., 2019). And several past methods in this area have conclusively demonstrated that, given sufficient number of queries, it is possible to achieve similarly effective attacks in the black-box setting akin to the white-box setting. However, as has also been demonstrated by past work Ilyas et al. (2019; 2018), the efficiency of these black-box attacks (the number of queries need to find an adversarial example) is fundamentally limited unless they can exploit the spatial correlation structure inherent in the model's gradients. Yet, at the same time, most previous methods have used rather ad-hoc methods of modeling such correlation structure, such as using "tiling" bases and priors over time Ilyas et al. (2019) that require attack vectors be constant over large regions, or by other means such as using smoothly-varying perturbations Ilyas et al. (2018) to estimate these gradients.

In this work, we present a new, more rigorous approach to model the correlation structure of the gradients within the black-box adversarial setting. In particular, we propose to model the gradient of the model loss function with respect to the input image using a Gaussian Markov Random Field (GMRF). This approach offers a number of advantages over prior methods: 1) it naturally captures the spatial correlation observed empirically in most deep learning models; 2) using the model, we are able to compute exact posterior estimates over the true gradient given observed data, while also fitting the parameters of the GMRF itself via an expectation maximization (EM) approach; and 3) the method provides a natural alternative to uniformly sampling perturbations, based upon the eigenvectors of the covariance matrix. Although representing the joint covariance over the entire input image may seem intractable for large-scale images, we can efficiently compute necessary terms for very general forms of grid-based GMRFs using the Fast Fourier Transform (FFT).

We evaluate our approach by attempting to find adversarial examples, over multiple different data sets and model architectures, using the GMRF combined with a very simple greedy zeroth order search technique; the method effectively forms a "black-box" version of the fast gradient sign method (FGSM), by constructing an estimate of the gradient at the input image itself, then taking a single signed step in this direction. Despite its simplicity, we show that owing to the correlation structure provided by the GMRF model, the approach outperforms more complex approaches such as the BANDITS-TD (Ilyas et al., 2019) or PARSIMONIOUS (Moon et al., 2019) methods (the current state of the art in black-box attacks), especially for small query budgets.

## 2 RELATED WORK

Black-box adversarial attacks can be broadly categorized across a few different dimensions: optimization-based versus transfer-based attacks, and score-based versus decision-based attacks.

In the optimization-based adversarial setting, the adversarial attack is formulated as the problem of maximizing some loss function (e.g., the accuracy of the classifier or some continuous variant) using a zeroth order oracle, i.e., by making queries to the classifier. And within this optimization setting, there is an important distinction between score-based attacks, which directly observe a traditional model loss, class probability, or other continuous output of the classifier on a given a example, versus decision-based attacks, which only observe the hard label predicted by the classifier. Decision based attacks have been studied by Brendel et al. (2017); Chen et al. (2019; 2017), and (not surprisingly) typically require more queries to the classifier than the score-based setting.

In the regime of score-based attacks, the first such iterative attack on a class of binary classifiers was first studied by Nelson et al. (2012). A real-world application of black-box attacks to fool a PDF malware classifier was demonstrated by (Xu et al., 2016), for which a genetic algorithm was used. Narodytska & Kasiviswanathan (2017) demonstrated the first black-box attack on deep neural networks. Subsequently black-box attacks based on zeroth order optimization schemes, using techniques such as KWSA (Kiefer et al., 1952) and RDSA (Nesterov & Spokoiny, 2017) were developed in Chen et al. (2017); Ilyas et al. (2018). Though Chen et al. (2017) generated successful attacks attaining high attack accuracy, the method was found to be extremely query hungry which was then remedied to an extent by (Ilyas et al., 2018). In Ilyas et al. (2019), the authors exploit correlation of gradients across iterations by setting a prior and use a piece wise constant perturbation, i.e., tiling to develop a query efficient black-box method. Recently Moon et al. (2019) used a combinatorial optimization perspective to address the black-box adversarial attack problem.

A concurrent line of work Papernot et al. (2017) has considered the transfer-based setting, rather than the optimization setting. These approaches create adversarial attacks by training a surrogate network with the aim to mimic the target model's decisions, which are then obtained through black-box queries. With the substitute model in place, the attack method then uses white-box attack strategies in order to transfer the attacks to the original target model. However, substitute network based attack strategies have been found to have a higher query complexity than those based on gradient estimation.

The exploitation of the structure of the input data space so as to append a regularizer has been recently found to be effective for robust learning. In particular, in Lin et al. (2019) showed that by using Wasserstein-2 geometry to capture semantically meaningful neighborhoods in the space of images helps to learn discriminative models that are robust to in-class variations of the input data.

**Setting of this work**   In this paper, we are specifically focused on the optimization-based, score-based setting, following most directly upon the work of (Chen et al., 2017; Ilyas et al., 2018; 2019; Moon et al., 2019). However, our contribution is also largely orthogonal to the methods presented in these prior works. Specifically, we show that by modeling the covariance structure of the gradient using a Gaussian MRF, a very simple approach (which largely mirrors the simple black box search from (Ilyas et al., 2018)) achieves performance that is competitive than the best current methods, especially when using relatively few queries. We further emphasize that while we focus on this simple search strategy here, nothing would prevent this GMRF approach from being applied to other black-box search strategies as well.

## 3 ADVERSARIAL ATTACKS

In the context of classifiers, adversarial examples are carefully crafted inputs to the classifier which have been perturbed by an additive perturbation so as to cause the classifier to misclassify the input. In particular, so as to ensure minimal visual distortion, the perturbation is subjected to a constraint in terms of its magnitude typically pre-specified in terms of $\ell_p$-norm, for some fixed $p$, less than some $\epsilon_p$. Furthermore in the context of classifiers, attacks can be further classified into targeted or untargeted attacks. For simplicity and brevity, in this paper, we restrict our attention to untargeted attacks.

Formally, define a classifier $C : \mathcal{X} \mapsto \mathcal{Y}$ with a corresponding classification loss function $L(\mathbf{x}, y)$, where $\mathbf{x} \in \mathcal{X}$ is the input to the classifier, $y \in \mathcal{Y}$, $\mathcal{X}$ is the set of inputs and $\mathcal{Y}$ is the set of labels. Technically speaking, the objective of generating a misclassified example can be posed as

an optimization problem. In particular, the aim is to generate an adversarial example $\mathbf{x}'$ for a given input $\mathbf{x}$ which maximizes $L(\mathbf{x}', y)$ but still remains $\epsilon_p$-close in terms of a specified metric, to the original input. Thus, the generation of an adversarial attack can be formalized as a constrained optimization as follows:

$$\mathbf{x}' = \underset{\mathbf{x}':\|\mathbf{x}'-\mathbf{x}\|_p \leq \epsilon_p}{\arg\max} L(\mathbf{x}', y). \tag{1}$$

We give a brief overview of adversarial attacks categorized in terms of access to information namely, white-box and black-box attacks.

### 3.1 WHITE-BOX ADVERSARIAL ATTACKS

White-box settings assume access to the entire classifier and the analytical form of the possibly non-convex classifier loss function. White-box methods can be further categorized into single iteration and multiple iterations based methods. In the class of single iteration white-box methods, the Fast Gradient Sign Method (FGSM) has been very successful, which computes the adversarial input in the following way:

$$\mathbf{x}' = \mathbf{x} + \epsilon_p \text{sign} \left( \nabla L(\mathbf{x}, y) \right). \tag{2}$$

FGSM is however limited to generation of $\ell_\infty$ based bounded adversarial inputs.

Given, a constrained optimization problem at hand with access to a first order oracle, the most effective method is *projected gradient descent (PGD)*. This multi iteration method generates the adversarial input $\mathbf{x}_k$ by performing $k$ iterations with $\mathbf{x}_0 = \mathbf{x}$, where $k$ is specified apriori. In particular, at the $l$-th iteration, PGD generates the perturbed input $\mathbf{x}_l$ as follows:

$$\mathbf{x}_l = \Pi_{B_p(\mathbf{x}, \epsilon)}(\mathbf{x}_{l-1} + \eta \mathbf{s}_l) \qquad \text{with } \mathbf{s}_l = \Pi_{\partial B_p(0,1)} \nabla_x L(\mathbf{x}_{l-1}, y), \tag{3}$$

where $\Pi_S$ denotes the projection onto the set $S$, $B_p(\mathbf{x}', \varepsilon')$ is the $\ell_p$ ball of radius $\varepsilon'$ centered at $\mathbf{x}'$, $\eta$ denotes the step size, and $\partial U$ is the boundary of a set $U$. By making, $\mathbf{s}_l$ to be the projection of the gradient $\nabla_x L(\mathbf{x}_{l-1}, y)$ at $x_{l-1}$ onto the unit $\ell_p$ ball, it is ensured that $\mathbf{s}_l$ is the unit $\ell_p$-norm vector that has the largest inner product with $\nabla_x L(\mathbf{x}_{l-1}, y)$. When $p = 2$, the projection corresponds to the normalized gradient, while for $p = \infty$, the projection corresponds to the sign of the gradient. Moreover, due to the projection at each iteration, the adversarial input generated at every iteration conforms to the specified constraint.

However, in most real world deployments, it is impractical to assume complete access to the classifier and analytic form of the corresponding loss function, which makes black-box settings more realistic.

### 3.2 BLACK-BOX ADVERSARIAL ATTACKS

In a typical *black-box* setting, the adversary has only access to a zeroth order oracle, which when queried for an input $(\mathbf{x}, y)$, yields the value of the loss function $L(\mathbf{x}, y)$. In spite of the information constraints and typically high dimensional inputs, black-box attacks have been shown to be pretty effective (Ilyas et al., 2019; 2018; Moon et al., 2019).

The main building block of black-box methods is finite difference schemes so as to estimate gradients. Two of the most widely finite difference schemes are the Kiefer-Wolfowitz Stochastic Approximation (KWSA)Kiefer et al. (1952) and Random Directions Stochastic Approximation (RDSA)(Nesterov & Spokoiny, 2017). KWSA operates as follows:

$$\widehat{\nabla}_x L(\mathbf{x}, y) = \sum_{k=1}^{d} \mathbf{e}_k \left( L(\mathbf{x} + \delta \mathbf{e}_k, y) - L(\mathbf{x}, y) \right) / \delta \approx \sum_{k=1}^{d} \mathbf{e}_k \nabla_x L(\mathbf{x}, y) \mathbf{e}_k, \tag{4}$$

where $\mathbf{e}_1, \ldots, \mathbf{e}_d$ are canonical basis vectors. The estimator can be further extended to higher order finite difference operators, but in the face of possibly non-smooth loss functions do not improve the accuracy of the estimator at the cost of additional queries. Hence, the first or the second order finite difference operators have been proven to be extremely effective. Though KWSA yields reasonably accurate gradient estimates, it is prohibitively query hungry. For instance, for Inception-v3 classifier for ImageNet, for every gradient estimate KWSA would require $299 \times 299 \times 3$ queries. RDSA provides a better alternative which operates as follows:

$$\widehat{\nabla}_x L(\mathbf{x}, y) = \frac{1}{m} \sum_{k=1}^{m} \mathbf{z}_k \left( L(\mathbf{x} + \delta \mathbf{z}_k, y) - L(\mathbf{x}, y) \right) / \delta, \tag{5}$$

where $\mathbf{z}_k$'s are usually drawn from a normal distribution. While RDSA is more query efficient than KWSA, it still needs a lot of queries to have a reasonably accurate estimate which typically scales with dimension. The step size $\delta > 0$ in RDSA and KWSA is a key parameter of choice; a higher $\delta$ could lead to extremely biased estimates, while a lower $\delta$ can lead to an unstable estimator. In light of the two aforementioned gradient estimation schemes, the PGD attack (c.f. equation 3) can be suitably modified to suit black-box attacks as follows:

$$\mathbf{x}_l = \Pi_{B_p(\mathbf{x},\epsilon)}(\mathbf{x}_{l-1} + \eta \widehat{\mathbf{s}}_l) \qquad \text{with} \ \ \widehat{\mathbf{s}}_l = \Pi_{\partial B_p(0,1)} \widehat{\nabla}_x L(\mathbf{x}_{l-1}, y). \qquad (6)$$

However, owing to the biased gradient estimates, though a PGD based black-box attack is successful turns out to be query hungry. In particular, in order to ensure sufficient increase of the objective at each iteration the query complexity scales with dimension and hence is prohibitively large. However, most if not all successful black-box adversarial attacks tend to be multi iteration based methods. In the sequel, we develop a query efficient single step black-box adversarial attack.

# 4 QUERY EFFICIENT SINGLE ITERATION BLACK-BOX ATTACKS

In this section, we develop the query efficient single iteration black-box attack method.

## 4.1 GRADIENT CORRELATION

In most black-box adversarial attacks, the gradient terms across different images are implicitly assumed to be independent from each other. However, even inspecting adversarial examples visually, it is apparent that the gradients across different images exhibit correlation. In fact, gradient terms seem to be heavily correlated, and a black-box method aiming to find adversarial examples using as few queries as possible should exploit this correlation. We propose to exploit and model these correlations using a Gaussian Markov random field. Formally, letting $\mathbf{x}$ be the input to a classifier, and $\mathbf{g} = \nabla L(\mathbf{x}, y)$ the gradient of the loss function with respect to the input, we are attempting to query and estimate the gradient, then we aim to put a prior distribution over $\mathbf{g}$

$$\mathbf{g} \sim \mathcal{N}(0, \Sigma) \qquad (7)$$

where $\Sigma$ is a non-identity covariance matrix modeling the correlation between terms. Following common practice we are not going to model Sigma, but rather model the inverse covariance matrix $\Lambda = \Sigma^{-1}$, a setting also known as the Gaussian Markov random field (GMRF) setting, given that the non-zero entries in $\Lambda$ correspond exactly to the edges in a graphical model describing the distribution. And even more specifically, we are not going to attempt to model each entry of $\Lambda$ separately, but use a parameterized Gaussian MRF with relatively few free parameters. For example, if $\mathbf{x}$ is a 2D image, then we may have one parameter $\alpha$ governing the diagonal terms $\Lambda_{i,i} = \alpha, \forall i$, and another governing adjacent pixels $\Lambda_{i,j} = \beta$ for $i, j$ corresponding to indices that are neighboring in the original image. We will jointly refer to all the parameters of this model as $\theta$, so in this case $\theta = (\alpha, \beta)$, and we refer to the resulting $\Lambda$ as $\Lambda(\theta)$.

We then consider the problem of fitting a parameterized MRF model to estimate gradients of inputs $\mathbf{x}^{(1)}, \ldots, \mathbf{x}^{(m)}$, using $n$ directional derivatives for each input given by $\mathbf{G} = [\mathbf{g}^{(1)}, \ldots, \mathbf{g}^{(mn)}]$. The maximum likelihood estimation for this problem is precisely the optimization problem

$$\min_{\theta} \ \operatorname{tr}(\mathbf{S}\Lambda(\theta)) - \operatorname{logdet}(\Lambda(\theta)), \qquad (8)$$

where $\mathbf{S} = \frac{1}{mn} \sum_i \mathbf{g}^{(i)} \mathbf{g}^{(i)\top}$ is the sample covariance and $\operatorname{logdet}$ denotes the log determinant. This is in fact the standard Gaussian maximum likelihood estimation problem. While this is a standard problem for the case of general covariance (minimizing $\Lambda$ is just the inverse of the same covariance), when we use a parameterized form of $\Lambda$, it becomes less clear how to solve this optimization problem efficiently. As we show, however, this optimization problem can be easily solved using the Fourier Transform; we focus for simplicity of presentation on the 2D case, but the method is easily generalizable to three dimensional convolutions to capture color channels in addition to the spatial dimensions itself (and we use this 3D form for all color images). First, we focus on evaluating the

trace term. The key idea here is that the Lambda operator can be viewed as a (circular) convolution[1]

$$\mathbf{K} = \begin{bmatrix} 0 & \beta & 0 \\ \beta & \alpha & \beta \\ 0 & \beta & 0 \end{bmatrix},$$

which then lets us compute $\mathrm{tr}(S\Lambda(\theta))$ as sum of the elements of the product of $\mathbf{G}$ and the zero padded 2D convolution of $\mathbf{G}$ and $\mathbf{K}$. For the log determinant term, we can again exploit the fact that $\Lambda$ is a convolution operator. Specifically, because it is a convolution, we know it can be diagonalized using the discrete Fourier transform.

$$\Lambda = \mathbf{Q}^H \mathbf{D} \mathbf{Q}$$

where $\mathbf{Q}$ is the the Fast Fourier Transform (FFT) basis, and the eigenvalues being the diagonal elements of $\mathbf{D}$ can be found by an Fast Fourier Transform (FFT) to the zero-padded convolution operator; thus, we can compute the log determinant term by simply taking the sum of the log of the FFT-computed eigenvalues. We then employ Newton's method to optimize the objective. The entire procedure of estimating the GMRF parameters is depicted in Algorithm 1. For a $N \times N$ sized image, the dominating cost for the procedure will be the $O(N^2 \log N)$ computation of the 2D FFT; this constrasts with the $O(N^6)$ naive complexity of forming, e.g., the naive eigen decomposition of the $N^2 \times N^2$ inverse covariance.

---

**Algorithm 1** Solving for GMRF

---

1: **procedure** SOLVING GMRF($\{\mathbf{x}^{(i)}\}_{i=1}^m, \delta$)
2:    Draw $n$ vectors $\mathbf{u}^{(1)}, \ldots, \mathbf{u}^{(n)} \sim \mathcal{N}(0, \mathbf{I})$
3:    Estimate $\mathbf{g}^{(1)}, \ldots, \mathbf{g}^{(mn)}$, where $\mathbf{g}^{ij} = \mathbf{u}^{(j)} \left( L(\mathbf{x}^{(i)} + \delta\mathbf{u}^{(j)}, y) - L(\mathbf{x}^{(i)} - \delta\mathbf{u}^{(j)}, y) \right)/2\delta$
4:    Generate $\widehat{\mathbf{G}}$ from $\mathbf{G}$ by concatenating along each dimension.
5:    Calculate $\mathrm{tr}(S\Lambda(\theta)) = \mathrm{sum}\left( \mathbf{G} \times \mathrm{conv2d}(\mathbf{K}, \widehat{\mathbf{G}}) \right)$; conv2d denotes 2D convolution
6:    Calculate $\mathrm{logdet}(\Lambda(\theta)) = \mathrm{sum}\left( \log(\mathrm{FFT}(\mathbf{A})) \right)$; $\mathbf{A}$ is the zero-padded convolution operator
7:    Use Newton's method to minimize the objective in equation 8
8:    **return** $\theta$

---

### 4.2 GRADIENT ESTIMATION

Under the aforementioned GMRF framework, we can interpret black-box gradient estimation as a Gaussian inference problem. Specifically, in our setting above, we have assumed that the gradient at a point $\mathbf{x}$ follows the normal distribution with the prescribed inverse covariance

$$\mathbf{g} \sim \mathcal{N}(0, \Lambda^{-1}).$$

When we observe the loss function value at some point $\mathbf{x}'$, this can be viewed as a noisy observation of the gradient

$$L(\mathbf{x}') \approx L(\mathbf{x}) + \mathbf{g}^\top (\mathbf{x}' - \mathbf{x}),$$

where we do an abuse of notation by dropping $y$ in $L(\mathbf{x}, y)$. Thus, given a set of sample points $\mathbf{x}^{(1)}, \ldots, \mathbf{x}^{(m)}$ and their corresponding loss function values $L(\mathbf{x}^{(1)}), \ldots, L(\mathbf{x}^{(m)})$, we have the following characterization of the distribution

$$\mathbf{L}_1 | \mathbf{g} \sim \mathcal{N}(\mathbf{X}\mathbf{g}, \sigma^2 \mathbf{I}),$$

where

$$\mathbf{L}_1 = \left[ L(\mathbf{x}^{(1)}) - L(\mathbf{x}), \cdots, L(\mathbf{x}^{(m)}) - L(\mathbf{x}) \right], \quad \mathbf{X} = \left[ (\mathbf{x}^{(1)} - \mathbf{x})^\top, \cdots, (\mathbf{x}^{(m)} - \mathbf{x})^\top \right].$$

The perturbed points $\{\mathbf{x}'^{(i)}\}_{i=1}^m$ are generated according to supplied vectors $\{\mathbf{z}^{(1)}\}_{i=1}^m$ to the procedure. Under this condition, the posterior $\mathbf{g} | \mathbf{L}_1$ is given by

$$\mathbf{g} | \mathbf{L}_1 \sim \mathcal{N}\left( \left( \Lambda + \mathbf{X}^\top \mathbf{X}/\sigma^2 \right)^{-1} \mathbf{X}^\top \mathbf{L}_1/\sigma^2, \left( \Lambda + \mathbf{X}^\top \mathbf{X}/\sigma^2 \right)^{-1} \right).$$

---

[1]The FFT operation technically operates circular convolutions (meaning the convolution wraps around the image), and thus the covariance naively models a correlation between, e.g., the first and last rows of an image. However, this is a minor issue in practice since: 1) it can be largely mitigated by zero-padding the input image before applying the FFT-based convolution, and 2) even if ignored entirely, the effect of a few additional circular terms in the covariance estimation is minimal.

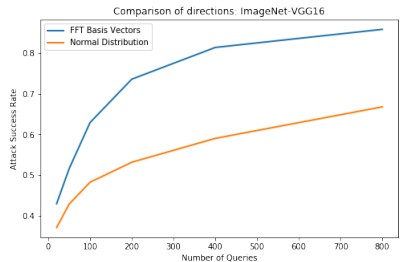

Figure 1: Summary of comparison of attack accuracy with query budgets for FFT basis vectors and normal distribution for $\ell_\infty$ with $\epsilon = 0.05$ on VGG16-bn trained on ImageNet. attacks

The matrix of interest that we need to solve for is the inverse covariance term

$$\Lambda + \frac{1}{\sigma^2}\mathbf{X}^\top\mathbf{X}.$$

This is a convolution plus a low rank matrix (the $\mathbf{X}^\top\mathbf{X}$ term is rank $m$, and we typically have $m \ll n$ because we have relatively few samples and a high dimensional input). We can not solve for this matrix exactly using the FFT, but we can still solve for it efficiently (requiring only an $m \times m$ inverse) using the matrix inversion lemma, specifically using Woodburry's matrix inversion lemma

$$\left(\Lambda + \frac{1}{\sigma^2}\mathbf{X}^\top\mathbf{X}\right)^{-1} = \Lambda^{-1} - \Lambda^{-1}\mathbf{X}^\top\left(\sigma^2\mathbf{I} + \mathbf{X}\Lambda^{-1}\mathbf{X}^\top\right)^{-1}\mathbf{X}\Lambda^{-1}.$$

Since the term needs to be computed explicitly for at least the inner inverse, we explicitly maintain the term $\mathbf{U} = \Lambda^{-1}\mathbf{X}^\top$. Note that in the sequential sampling setting (where we sequentially sample $\mathbf{x}^{(i)}$ points one at a time), this matrix could be maintained over all samples, so that just a single solve would be required for each new sample. We use the mean of the the conditional distribution $\mathbf{g}|\mathbf{L}_1$ as the gradient estimate. The gradient estimation procedure is depicted in Algorithm 2

---

**Algorithm 2** Gradient Estimation

---

1: **procedure** GRADEST($\mathbf{x}, \{\mathbf{z}^{(i)}\}_{i=1}^m, \sigma, \delta_1, \theta$)
2:     Compute $\mathbf{L}_1$ by querying the model at $\{\mathbf{x} + \delta_1\mathbf{z}^{(i)}\}_{i=1}^m$ and $\{\mathbf{x} - \delta_1\mathbf{z}^{(i)}\}_{i=1}^m$
3:     Compute $\mathbf{X} = 2\delta_1[\mathbf{z}^{(1)}, \ldots, \mathbf{z}^{(m)}]$
4:     Compute $\widehat{\mathbf{g}} = \left(\Lambda + \mathbf{X}^\top\mathbf{X}/\sigma^2\right)^{-1}\mathbf{X}^\top\mathbf{L}_1/\sigma^2$ using FFT
5:     **return** $\widehat{\mathbf{g}}$

---

### 4.3   GRADIENT ESTIMATION: DIRECTIONAL DERIVATIVES

In order to estimate the gradient efficiently in Algorithm 2, a key role is played by the vectors $\{\mathbf{z}^{(1)}\}_{i=1}^m$ which perturb the input. One particular choice being sampling the directions from a normal distribution. However, it is worth noting that the inverse covariance distribution of the gradients by construction is a convolution operator and hence is diagonalized by the FFT basis. In particular, the low frequency components of the FFT basis vectors enhanced the attack accuracy significantly, an example of which is depicted in Figure 1 for black-box attacks on VGG16-bn classifier for ImageNet with $\epsilon = 0.05$. With the gradient estimate at hand, the adversarial input for $\mathbf{x}$ is generated using FGSM as follows:

$$\mathbf{x}_{adv} = \mathbf{x} + \epsilon\text{sign}\left(\widehat{\mathbf{g}}\right). \tag{9}$$

The entire procedure consisting of GMRF inference and gradient inference is presented in Algorithm 3. We provide more details about the performance of our gradient estimation scheme in terms of various metrics such as mean square error and cosine similarity between the estimated gradient and the true gradients in the Appendix A.6.

---

**Algorithm 3** GMRF based black-box FGSM

---
1: **procedure** BB-FGSM$(\sigma, \delta_1, \delta, \epsilon)$
2: $\quad \theta \leftarrow$ SOLVING GMRF$(\{\mathbf{x}^{(i)}\}_{i=1}^{m}, \delta)$
3: $\quad \{\mathbf{z}^{(1)}\}_{i=1}^{m} \leftarrow$ FFT basis vectors
4: $\quad \widehat{\mathbf{g}} \leftarrow$ GRADEST$(\mathbf{x}, \{\mathbf{z}^{(i)}\}_{i=1}^{n}, \sigma, \delta_1, \theta)$
5: $\quad \mathbf{x}_{adv} = \mathbf{x} + \epsilon\,\text{sign}\,(\widehat{\mathbf{g}})$
6: $\quad$ **return** $\mathbf{x}_{adv}$

---

## 5 EXPERIMENTS

In this section, we focus on the untargeted attack setting where the goal is to generate a perturbation so as to get an original image originally classified correctly by the classification model to be misclassified to any class other than the original class. In particular, we consider $\ell_\infty$ attacks on ImageNet (Deng et al., 2009) and MNIST (Lecun et al., 1998) and evaluate the performance in terms of attack success rate with a given query budget and average query count.

The attack success rate is defined as the ratio of the number of images successfully misclassified to the total number of input images. Among the set of input images, we discard images which were misclassified by the classifier. The average query count is computed on the queries made for successful attacks.

### 5.1 EXPERIMENTS ON MNIST

We compare the performance of the proposed method with that of white-box FGSM (Goodfellow et al., 2014) across different values of $\ell_\infty$ bounds ranging from 0.05 to 0.3 in increments of 0.05 over query budgets from 20 to 200. We use the pre-trained LeNet model available from Pytorch to demonstrate the attacks. We use all the correctly classified images from the 10,000 images (scaled to [0, 1]) in the MNIST test set. To generate the sample gradients, for GMRF estimation, $1,000$ queries were used. To further illustrate the importance of incorporating a non-identity gradient covariance, we also provide experimental results for a version of our proposed algorithm which takes the gradient covariance to be an identity matrix. As shown in Figure 2, our proposed attack method exhibits better attack accuracy than white-box FGSM in around 75 queries and the gap in the performance is further magnified with increasing number of queries. On the other hand, the version of our algorithm with identity gradient covariance consistently under performs with respect to the white-box FGSM attack. In particular, the identity covariance version only reaches close to the white-box FGSM attack performance in 200 queries. This further illustrates the effectiveness of our proposed algorithm and the importance of modelling the gradient covariance as a GMRF and more generally as a non-identity covariance matrix.

The superior performance of our proposed framework as compared to white-box FGSM as demonstrated in Figure 2 can be attributed to the following reason. First, incorporating the gradient non-identity covariance structure into the gradient estimation scheme, allows our perturbation to be able to use structural gradient information from other images too. On the other hand, white box FGSM treats gradient of every image to be independent of gradients of other images. This is further illustrated by our experimental findings based on the version of our algorithm considering the gradient covariance to be an identity matrix.

### 5.2 EXPERIMENTS ON IMAGENET

We compare the performance of the proposed method with that of NES (Ilyas et al., 2018), BANDITS-TD (Ilyas et al., 2019) and PARSIMONIOUS (Moon et al., 2019), which are the current state of the art in $\ell_\infty$ based black-box attacks. For ImageNet, we consider three classifiers namely, ResNet50 (He et al., 2015), Inception-v3 (Szegedy et al., 2015) and VGG16-bn (Simonyan & Zisserman, 2014). We use the pre-trained models provided by PyTorch for attacking these classifiers. We use all the correctly classified images from the 50,000 images (scaled to [0, 1]) in the ImageNet validation set.

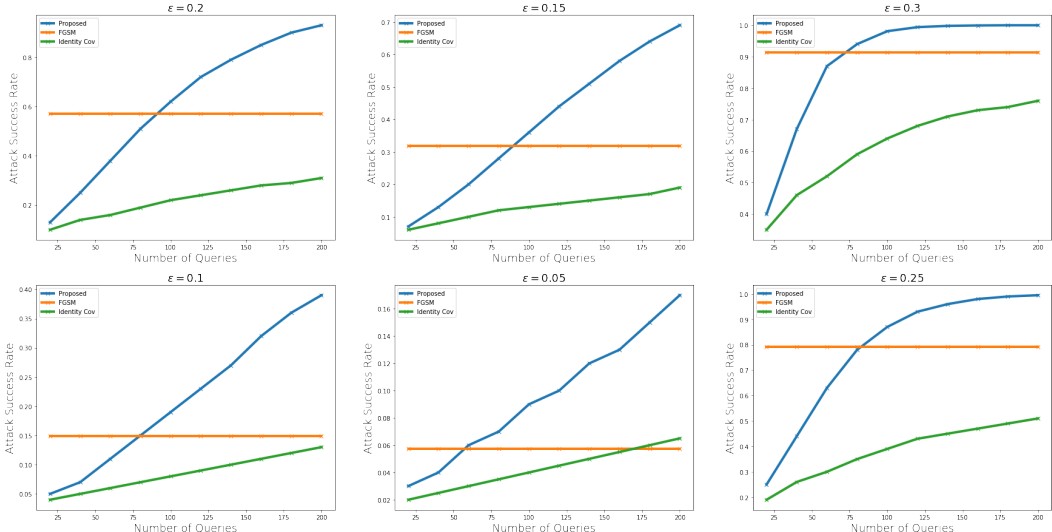

Figure 2: Summary of comparison of the proposed attack and the proposed attack with identity gradient covariance with white-box FGSM for attack accuracy with query budgets for MNIST on LeNet

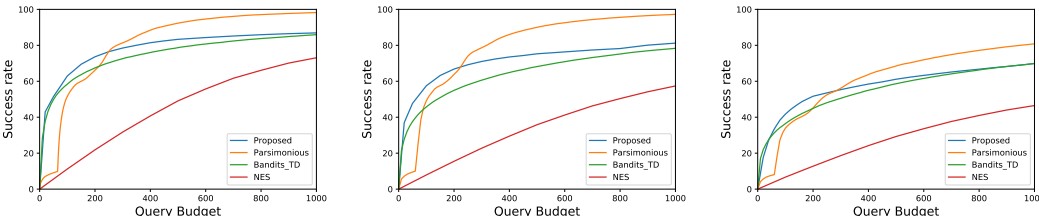

Figure 3: VGG16-bn: Attack success rate as a function of query budget

Figure 4: ResNet50: Attack success rate as a function of query budget

Figure 5: Inception-v3: Attack success rate as a function of query budget

The $\ell_\infty$ perturbation bound is set to $\epsilon = 0.05$. We use the implementation[2] and hyperparameters provided by Ilyas et al. (2019) for NES and BANDITS-TD. Similarly for PARSIMONIOUS, we use the implementation[3] and hyperparameters given by (Moon et al., 2019). The last 50 images of the ImageNet validation set are used for estimating the parameters of the GMRF. In particular, to generate the sample gradients, for each model, $5,000$ queries were used for the GMRF estimation. The specifics of the GMRF model, the values of the parameters and the associated hyperparameters (which were obtained by grid search) for the proposed algorithm for the three classifiers are relegated to the Appendix. Figures 3 - 5 show the evolution of attack accuracy with different querying budgets. In the regime of low query budget with less than 200 query budget, our algorithm outperforms PARSIMONIOUS, though it exhibits inferior performance in the higher query budget regime.

As shown in Table 1, our proposed algorithm in spite of being a single-step attack, outperforms BANDITS-TD and NES by achieving higher attack success rate. Despite the higher success rate, the proposed method uses fewer queries on an average as compared to BANDITS-TD and NES. Thus, the proposed method strictly dominates BANDITS-TD and NES on every metric.

The proficiency of our proposed scheme in the low query budget regime can be attributed to the utilization of the correlation between gradients across images and the usage of FFT basis vectors for calculating directional derivatives. In particular, the FFT basis vectors being the eigen vectors of the covariance matrix provide for systematic dimensionality reduction.

---

[2]https://github.com/MadryLab/blackbox-bandits
[3]https://github.com/snu-mllab/parsimonious-blackbox-attack

Table 1: Summary of $\ell_\infty$ attacks with $\epsilon = 0.05$ ImageNet attacks using NES, BANDITS-TD, PAR-SIMONIOUS and the proposed method with a query budget of 1000 per image

| Attack | VGG16-bn | | Resnet50 | | Inception-v3 | |
|---|---|---|---|---|---|---|
| | Success | Avg. Queries | Success | Avg. Queries | Success | Avg. Queries |
| NES | 73.08% | 441.38 | 57.73% | 471.24 | 46.75% | 467.02 |
| Parsimonious | 98.24% | 175.76 | 95.73% | 186.43 | 80.86% | 247.11 |
| Bandits$_{TD}$ | 85.91% | 133.67 | 78.80% | 181.15 | 69.83% | 212.55 |
| Proposed | 86.92% | 118.28 | 81.22% | 132.09 | 69.97% | 192.69 |

## 6 CONCLUSION

In this paper, we have developed a GMRF based covariance modeling technique so as to streamline the gradient estimation scheme catered towards black-box adversarial attacks. In particular, due to the streamlined gradient estimation scheme, we could alleviate the issue of random directional derivative searches which plagues every zeroth order optimization scheme due to biased gradient estimates. The gradient estimation scheme can be used in any gradient based black-box adversarial attack method to attain higher attack accuracies with lower query counts. Our method facilitates single iteration based query efficient black-box attacks which we demonstrated to be as potent as multi-step attacks on multiple architectures and datasets in terms of attack success rate. We also employed techniques from matrix analysis and FFT to make our attack computationally efficient. Our results open avenues for more effective covariance modeling techniques so as to further streamline gradient estimation schemes so as to facilitate more query efficient black-box adversarial attacks.

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

## A  APPENDIX

### A.1  MNIST EXPERIMENTS

The GMRF model used for MNIST is given by $\Lambda_{i,i} = \alpha, \Lambda_{i,i+1} = \Lambda_{i+1,i} = \Lambda_{i,i-1} = \Lambda_{i-1,i} = \beta, \Lambda_{i+1,i+1} = \Lambda_{i-1,i-1} = \Lambda_{i-1,i+1} = \Lambda_{i+1,i-1} = \gamma$, where $\Lambda_{i,j}$ denotes the $(i,j)$-th element of $\Lambda$. For estimating the GMRF parameters, we use the last 20 images of the MNIST test set and perturb each of them with 50 vectors drawn from a normal distribution. For the attack, we use low frequency basis vectors of the FFT basis. The following table gives the values of the different hyperparameters used in the attack. Except for the GMRF parameters, all the other parameters were determined using grid search.

### A.2  VGG16 EXPERIMENTS

The GMRF model used for VGG16 for Imagenet is given by $\Lambda_{0,i,i} = \alpha, \Lambda_{0,i,i+1} = \Lambda_{0,i+1,i} = \Lambda_{0,i,i-1} = \Lambda_{0,i-1,i} = \beta, \Lambda_{0,i+1,i+1} = \Lambda_{0,i-1,i+1} = \Lambda_{0,i-1,i-1} = \Lambda_{0,i+1,i-1} = \kappa, \Lambda_{1,i,i} = \Lambda_{-1,i,i} = \gamma$, where in $\Lambda_{k,i,j}$, $k$ denotes the channel. We also tried out GMRF models of lower and higher degree of association and we selected the one performing the best. For estimating the

Table 2: MNIST Experiment Settings

| | |
|---|---|
| $\delta$ | 0.1 |
| $\alpha$ | 21094408 |
| $\beta$ | $-5116365$ |
| $\gamma$ | 284558.1562 |
| $\sigma$ | $10^{-3}$ |
| $\delta_1$ | 0.15 |

GMRF parameters, we use the last 50 images of the ImageNet validation set and perturb each of them with 50 vectors drawn from a normal distribution. For the attack, we use low frequency basis vectors of the FFT basis. The following table gives the values of the different hyperparameters used in the attack. Except for the GMRF parameters, all the other parameters were determined using grid

Table 3: ImageNet VGG-16 Experiment Settings

| | |
|---|---|
| $\delta$ | 0.1 |
| $\alpha$ | 633.44 |
| $\beta$ | $-24.05$ |
| $\gamma$ | $-232.04$ |
| $\kappa$ | $-2.00$ |
| $\sigma$ | 1.0 |
| $\delta_1$ | 0.05 |

search.

## A.3   RESNET50 EXPERIMENTS

The GMRF model used for VGG16 for Imagenet is given by $\Lambda_{0,i,i} = \alpha, \Lambda_{0,i,i+1} = \Lambda_{0,i+1,i} = \Lambda_{0,i,i-1} = \Lambda_{0,i-1,i} = \beta, \Lambda_{0,i+1,i+1} = \Lambda_{0,i-1,i+1} = \Lambda_{0,i-1,i-1} = \Lambda_{0,i+1,i-1} = \kappa, \Lambda_{0,i,i+2} = \Lambda_{0,i,i-2} = \Lambda_{0,i-2,i} = \Lambda_{0,i+2,i} = \Lambda_{0,i+1,i+2} = \Lambda_{0,i-1,i+2} = \Lambda_{0,i+2,i+1} = \Lambda_{0,i+2,i-1} = \Lambda_{0,i-1,i-2} = \Lambda_{0,i+1,i-2} = \Lambda_{0,i-2,i-1} = \Lambda_{0,i-2,i+1} = \nu, \Lambda_{1,i,i} = \Lambda_{-1,i,i} = \gamma$, where in $\Lambda_{k,i,j}$, $k$ denotes the channel. We also tried out GMRF models of lower and higher degree of association and we selected the one performing the best. For estimating the GMRF parameters, we use the last 50 images of the ImageNet validation set and perturb each of them with 50 vectors drawn from a normal distribution. For the attack, we use low frequency basis vectors of the FFT basis. The following table gives the values of the different hyperparameters used in the attack. Except for the

Table 4: ImageNet ResNet50 Experiment Settings

| | |
|---|---|
| $\delta$ | 0.1 |
| $\alpha$ | 2631.93 |
| $\beta$ | $-263.33$ |
| $\gamma$ | $-837.16$ |
| $\kappa$ | 6.78 |
| $\nu$ | 28.09 |
| $\sigma$ | 0.5 |
| $\delta_1$ | 0.05 |

GMRF parameters, all the other parameters were determined using grid search.

### A.4  INCEPTION V3 EXPERIMENTS

The GMRF model used for VGG16 for Imagenet is given by $\Lambda_{0,i,i} = \alpha, \Lambda_{0,i,i+1} = \Lambda_{0,i+1,i} = \Lambda_{0,i,i-1} = \Lambda_{0,i-1,i} = \beta$, $\Lambda_{0,i+1,i+1} = \Lambda_{0,i-1,i+1} = \Lambda_{0,i-1,i-1} = \Lambda_{0,i+1,i-1} = \kappa$, $\Lambda_{0,i,i+2} = \Lambda_{0,i,i-2} = \Lambda_{0,i-2,i} = \Lambda_{0,i+2,i} = \Lambda_{0,i+1,i+2} = \Lambda_{0,i-1,i+2} = \Lambda_{0,i+2,i+1} = \Lambda_{0,i+2,i-1} = \Lambda_{0,i-1,i-2} = \Lambda_{0,i+1,i-2} = \Lambda_{0,i-2,i-1} = \Lambda_{0,i-2,i+1} = \nu$, $\Lambda_{1,i,i} = \Lambda_{-1,i,i} = \gamma$, where in $\Lambda_{k,i,j}$, $k$ denotes the channel. We also tried out GMRF models of lower and higher degree of association and we selected the one performing the best. For estimating the GMRF parameters, we use the last 50 images of the ImageNet validation set and perturb each of them with 50 vectors drawn from a normal distribution. For the attack, we use low frequency cosine basis vectors of the FFT basis. The following table gives the values of the different hyperparameters used in the attack. Except for the

Table 5: ImageNet Inception v3 Experiment Settings

| | |
|---|---|
| $\delta$ | 0.1 |
| $\alpha$ | 8964.89 |
| $\beta$ | $-2960.87$ |
| $\gamma$ | $-841.13$ |
| $\kappa$ | 1155.66 |
| $\nu$ | 286.03 |
| $\sigma$ | 0.5 |
| $\delta_1$ | 0.05 |

GMRF parameters, all the other parameters were determined using grid search.

### A.5  EFFICIENT COMPUTATION OF FFT BASIS

We use the fact that the covariance and the inverse covariance matrix because of being convolutional operators are diagonalized by the FFT basis. Let us assume the image is of size $c \times h \times w$, where $c$,$h$ and $w$ denote the number of channels, height and width of the gradient. We define a tensor $S$ of zeros of size $c \times h \times w \times 2$, where the last dimension is to account for both the real and complex components. In order to generate the lowest frequency basis vector, we set the the first element of the tensor of the first channel, i.e, $S_{0,0,0,0} = 1$ and take the inverse FFT. This gives us the lowest cosine basis vector. We do the same for the other channels, by just setting the corresponding component to 1 and taking the inverse FFT. Setting, $S_{0,0,0,1} = 1$ and then taking the inverse FFT yields the lowest frequent sine component. In order to generate the low frequency components, we start from the beginning of a row and proceed along diagonally by incrementing the row and column index by one. At each entry of the tensor, we repeat it for every channel once at a time.

### A.6  GRADIENT ESTIMATION PERFORMANCE

We illustrate the performance of our gradient estimation scheme in this section through experiments on the MNIST and the ImageNet dataset using LeNet and VGG-16bn respectively. We use two metrics namely, mean squared error of the normalized estimated gradient with respect to the normalized true gradient and the cosine similarity between the estimated gradient and the true gradient. In order to perform our analysis, we use 500 data samples from the test set to estimate the gradient using the GMRF framework. First, we estimate the GMRF parameters as previously described in Algorithm 1 and then perform the MAP estimation for the gradient. For MNIST, we demonstrate the gradient estimation performance on two regimes, i.e., 40 query budget where our scheme is outperfomed by white-box FGSM and 200 query budget where the vice versa happens. As evident from Figures 6b and 7b, our scheme generates gradient estimates which have low MSE. For, the 40 query budget setting, the cosine similarity is centered around 0.3, while for the 200 query budget setting, the cosine similarity is centered around 0.2. Had our gradient estimates completely aligned with that of the true gradient as in white-box FGSM our performance would have been upper bounded by the performance of white-box FGSM. In essence, our scheme finds directions for adversarial perturbations, which in itself does not maximize the loss but is able to find a direction which leads to misclassification of the examples.

The gradient estimation performance for ImageNet using VGG16-bn is depicted in Figure 8. In order to perform our analysis, we sample 500 data samples from the ImageNet validation set to estimate the gradient using the GMRF framework. First, we estimate the GMRF parameters as previously described in Algorithm 1 and then perform the MAP estimation for the gradient. We specifically consider the query budget to be 200. Out of the 400 correctly classified images, whitebox FGSM and our proposed algorithm attain attack accuracies of 0.9268 and 0.7804 respectively. While, the gradient estimation performance in terms of MSE is impressive, the cosine similarity shows that the estimated gradient does not quite coincide directionally with the true gradient. The difference in the directions explains the inferior performance of our proposed scheme in this regime. It is worth noting that the dimension of the input data for VGG16-bn is 150528. From classical results in zeroth-order optimization it is well known that in a $d$-dimensional space, $O(d)$ queries are required to obtain a nearly bias-free gradient estimate. Our framework uses only 200 queries to estimate the gradient which resides in a 150528 dimensional space. In spite of the possible erroneous directional characteristic of our estimated gradient, it still manages to achieve a 0.78 success rate and outperforms BANDITS-TD and PARSIMONIOUS in the 200 query budget regime.

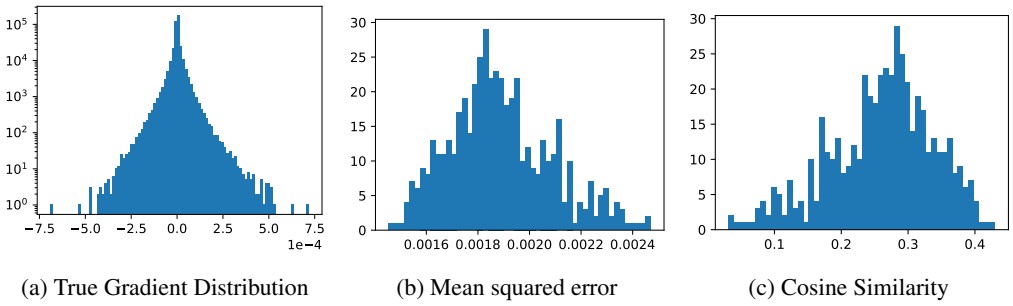

(a) True Gradient Distribution    (b) Mean squared error    (c) Cosine Similarity

Figure 6: LeNet, MNIST, 40 query budget

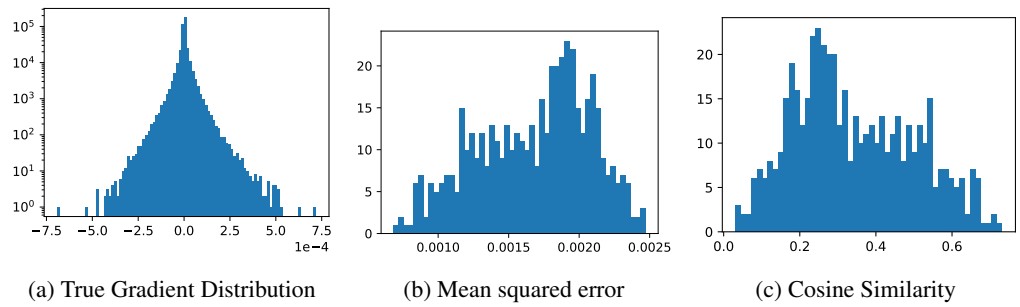

(a) True Gradient Distribution    (b) Mean squared error    (c) Cosine Similarity

Figure 7: LeNet, MNIST, 200 query budget

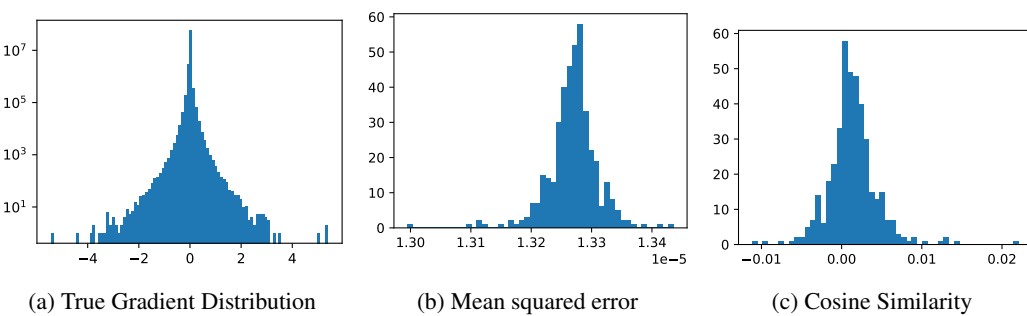

(a) True Gradient Distribution    (b) Mean squared error    (c) Cosine Similarity

Figure 8: VGG-16bn, ImageNet, 200 query budget

### A.7 AUTOCORRELATION

We provide further evidence for considering a non-identity covariance for modelling the gradient covariance. In figure 9, we plot the autocorrelation of the true gradients from 100 images sampled from the MNIST test set for two different kernel sizes, i.e., $9 \times 9$ and $11 \times 11$ for the LeNet model. In figure 10, we plot the autocorrelation of the third channel of the true gradients from 50 images sampled from the ImageNet validation set for two different kernel sizes, i.e., $9 \times 9$ and $11 \times 11$ for the VGG16-bn model. The autocorrelation plots show the correlation across dimensions and across images to be substantial so as to provide even more evidence and reason for the gradient model we have considered in this paper.

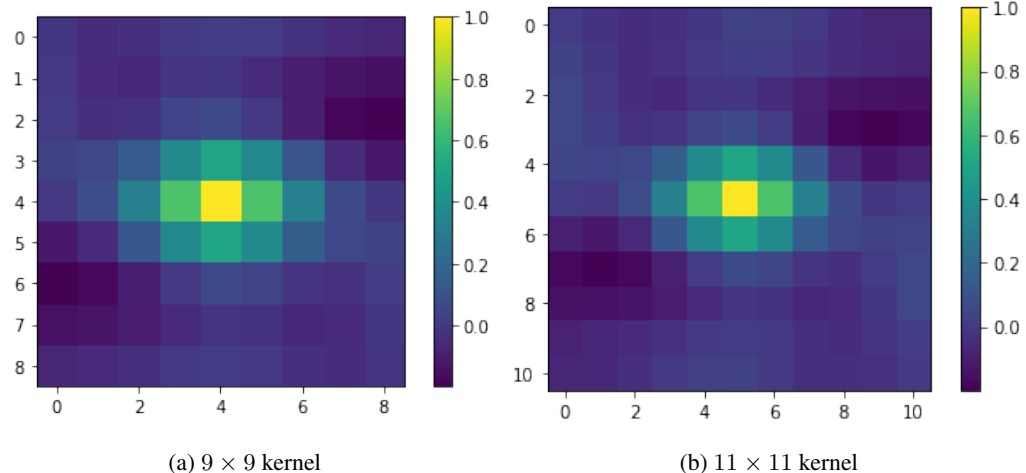

(a) $9 \times 9$ kernel             (b) $11 \times 11$ kernel

Figure 9: Autocorrelation of the gradients: MNIST, LeNet

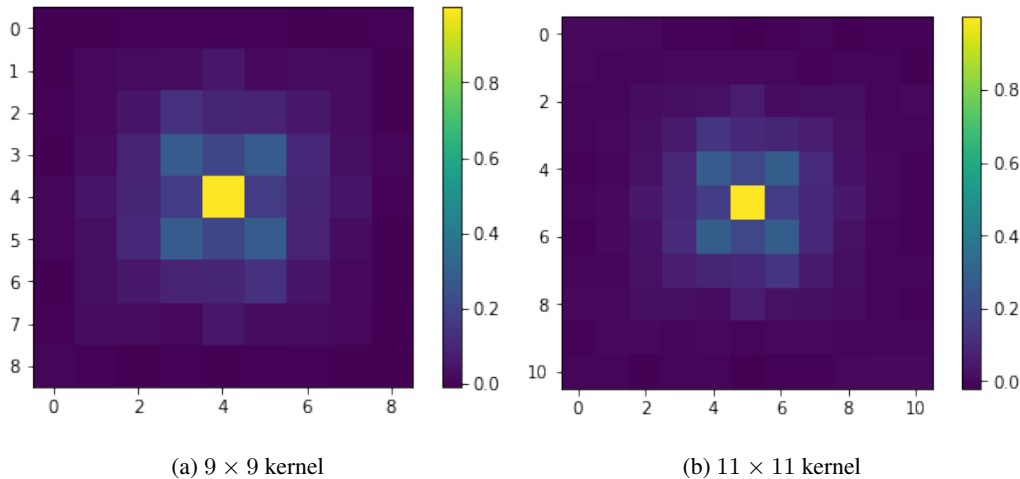

(a) $9 \times 9$ kernel             (b) $11 \times 11$ kernel

Figure 10: Autocorrelation of the gradients: ImageNet, VGG16-bn

