# OpenReview forum: "Gaussian MRF Covariance Modeling for Efficient Black-Box Adversarial Attacks"
_ICLR.cc/2020/Conference — Reject_

### Official Review · AnonReviewer1 · 2019-10-23
**Official Blind Review #1**

**Rating:** 8

**Review:**

This paper employs Markov random fields to exploit the input data structure and further model the covariance structure of the gradients. This embeds covariance structures of input data space into the gradient operator for an adversary attack.
They further use this gradient operator with a fast gradient sign Method. The numerics show effective for using fewer queries to obtain high attack accuracy. This paper is well written with clear derivations. I suggest the publication of the paper.

 In fact, modeling the structure of an input space structure into the adversary attack is a good direction. For similar intuitions in this direction, I recommend a related work:

         "A. Lin, Y. Dukler, W. Li, G. Montufar, Wasserstein Diffusion Tikhonov Regularization"


**Experience Assessment:**

I have published one or two papers in this area.

**Review Assessment: Checking Correctness Of Derivations And Theory:**

I assessed the sensibility of the derivations and theory.

**Review Assessment: Checking Correctness Of Experiments:**

I assessed the sensibility of the experiments.

**Review Assessment: Thoroughness In Paper Reading:**

I read the paper at least twice and used my best judgement in assessing the paper.

---

> ### Author Response · Authors · 2019-11-09
> **Response to Reviewer #1**
>
> Thank you for the review and for the encouraging and positive comments.  Additionally, thanks very much for pointing out this related work. We will add it and some discussion to the related work section in the revised draft.

---

### Official Review · AnonReviewer3 · 2019-10-23
**Official Blind Review #3**

**Rating:** 3

**Review:**

In this paper, the authors propose a method for black box adversarial image generation. The idea is to learn a parameterization of a precision matrix so that gradients of a network's loss are assumed to be drawn from a corresponding Gaussian. The parameters of this model are fit efficiently using the spectral theorem that their particular parameterization of the precision matrix allows them to use. Gradient estimation is then viewed as a Gaussian conditioning problem given observations (see last equation on page 5).

Overall, I think the method is elegant -- particular in comparison to many existing approaches in the literature that rely on highly complex machinery like genetic algorithms and the like. My main source of questions is the experimental results section, which I currently view as somewhat weak and a little confusing -- I would be more than happy to increase my score if my concerns are sufficiently addressed.

First, I'm not sure that the story told by Figure 3 and Table 1 is entirely clear. On the whole at a given sufficiently high success rate (say, 80%), it seems that the authors' approach consistently loses to the Parsimonious attack of Moon et al., 2019? The authors' method seems to perform strictly better than Bandits_{TD} and NES, but the gap between the Parsimonious attack and the authors' is quite substantial past 300 evaluations.

This ultimately leaves me with the question of what to do with this paper. The learning framework used to estimate gradients is clever, but doesn't seem to me to be methodologically groundbreaking to the point where it can stand on its own merits independent of its performance in comparison to other techniques. Particularly considering other papers have been published since the Ilyas et al., 2019 paper that outperform Bandits_{TD} and NES in terms of query efficiency, I worry that this paper simply presents a decent idea with middling results.

I'd therefore like the authors to primarily comment on the broader impact they believe their paper will have. The conclusion primarily focuses on the introduction of the method: does it have substantial merits past its empirical performance?

**Experience Assessment:**

I have published one or two papers in this area.

**Review Assessment: Checking Correctness Of Derivations And Theory:**

I carefully checked the derivations and theory.

**Review Assessment: Checking Correctness Of Experiments:**

I carefully checked the experiments.

**Review Assessment: Thoroughness In Paper Reading:**

I read the paper thoroughly.

---

> ### Author Response · Authors · 2019-11-09
> **Response to Reviewer #3**
>
> We thank the reviewer for the thorough review and the encouraging comments. The main takeaways of our framework are two folds:
>
> 1) Most attacks which involve zeroth order gradient estimation using classical finite difference methods resort to treat the gradient estimation of an image to be independent of the gradients of the past encountered images. To some extent, Bandits-TD alleviates this by considering a prior. However, by employing tiling in Bandits-TD the perturbations are locally smooth and conform to a very specific class of perturbations. Our methodology shows, by employing a simple yet effective GMRF framework to capture correlations between gradients yields remarkably good performance by just using a single step attack. Such a gradient covariance framework, can be added on top of any other multi-step methods and frameworks to yield potentially even more superior performance. We also provide a computationally efficient methodology to avoid high dimensional matrix inversions and expensive computations so as for our method to be seamlessly usable. Our methods opens up the question of coming up with more crafty covariance modeling techniques so as to enhance the performance of any black-box adversarial techniques.
>
> 2) Secondly, given the long line of zeroth order gradient estimation frameworks most black-box adversarial attacks primarily use vectors sampled from normal distribution or the canonical basis vectors for directional derivatives. Through our method, we particularly highlight the effectiveness of the usage of FFT basis vectors for directional derivatives. In particular through Figure 1, we highlight the remarkable increase in attack accuracy of using FFT basis vectors instead of vectors sampled from the normal distribution. Intuitively speaking, FFT basis vectors result in an inherent dimensionality reduction akin to the tiling step in Bandits-TD, while allowing for general perturbations as opposed to locally smooth perturbations as in Bandits-TD. Thus, we have reason to believe other black-box adversarial schemes could also immensely benefit from using FFT basis vectors for directional derivatives. We also provide a very simple framework to generate FFT basis vectors in a very computationally efficient manner.
>
> In summary, in addition to being a very simple one-step black-box attack which beats white box FGSM our method proposes use of gradient correlation across images and use of FFT basis vectors for directional derivatives which other methods would also benefit from.
>
> In regards to performance, while achieving higher attack accuracy is definitely desirable, it is also important to be query efficient. As our method especially does well in lower query budget regimes, it can be used as a baseline for other black-box adversarial attack schemes striving for query efficiency in the low query budget regime.

---

### Official Review · AnonReviewer2 · 2019-10-24
**Official Blind Review #2**

**Rating:** 3

**Review:**

This paper deals with the problem of finding an adversarial examples when only the output of a model can be evaluated, but not its gradient. The key idea of the paper is building a Gaussian MRF (a Gaussian with a sparse inverse covariance matrix with a special band structure) to maintain a model for the gradients for predicting search directions. The approach is sensible and uses the FFT trick applicable for diagonalizing covariance matrices with circulant structure.

- The ideas in this paper have practical utility. The paper is unfortunately not very carefully written, and the arguments require occasionally some guesswork.

- There is not a sufficient discussion of experimental findings. Why does the proposed method works better than a white box FGSM for instance?

- I don’t fully understand what ‘solving for GMRF’ means (Alg 1). I would expect it is estimating the parameters of the covariance function but this algorithm just calculates the likelihood.

- Given the simple coupling structure with parameter tying, this problem seems to be closely related to estimating AR(N) style models (alpha and beta parameters) so I am surprised to see only a general treatment.
In this problem, it seems much more natural to estimate theta and g recursively and concurrently, and there are very well known algorithms related to Kalman filtering. Please discuss.

- The evaluation of the idea is not complete While it is certainly interesting to see that such a simple heuristic can achieve comparable performance on standard datasets over other black-box attack methods in the limited query budget regime, I would expect to see some experiments that illustrate the quality of the gradient estimator more closely (not only its final effectiveness for finding a search direction inside of an optimization method) and more strong justification of the proposed model.

- As the entire contribution hinges on the observation that gradients across dimensions of an example as well as across examples in a dataset are correlated, it would have been also very informative to show estimates of autocorrelation functions of  the gradients on different datasets to justify the basic modelling choices.



**Experience Assessment:**

I have read many papers in this area.

**Review Assessment: Checking Correctness Of Derivations And Theory:**

I assessed the sensibility of the derivations and theory.

**Review Assessment: Checking Correctness Of Experiments:**

I assessed the sensibility of the experiments.

**Review Assessment: Thoroughness In Paper Reading:**

I read the paper at least twice and used my best judgement in assessing the paper.

---

> ### Author Response · Authors · 2019-11-09
> **Response to Reviewer #2**
>
> R: The ideas in this paper have practical utility. The paper is unfortunately not very carefully written, and the arguments require occasionally some guesswork.
> A: Thanks very much for the thorough review and the encouraging comments, and especially for pointing out which elements of the paper were unclear.  We will supplement our experimental findings with more discussions in a revised draft.  We’re currently running some experiments to address some of the points you raised, but given the timing we also wanted to respond to some of the points now, and will reply further over the remainder of the review period as the additional experiments are revisions are finishing.
>
> R: There is not a sufficient discussion of experimental findings. Why does the proposed method works better than a white box FGSM for instance?
> A: We will add further discussion for the experimental findings in the revised draft. We suspect that the superior performance of our proposed framework as compared to white box FGSM as demonstrated on the MNIST dataset can be attributed to two reasons.  First, incorporating the gradient non-identity covariance structure into the gradient estimation scheme, allows our perturbation to be able to use structural gradient information from other images too. On the other hand, white box FGSM treats gradient of every image to be independent of gradients of other images.  Second, the exact gradient of an image with respect to perturbations can often be a poor estimate of the actual “larger delta” gradient (i.e., the changes that actually result from larger delta perturbations).  For this reason, “large step” finite differencing can sometimes actually be a better direction when it comes to creating an adversarial example.  We’re working on some experiments that will hopefully clarify better which effect is dominant here, and while it may be a bit detailed for the main text, we could for instance include it in an appendix.
>
> R:  I don’t fully understand what ‘solving for GMRF’ means (Alg 1). I would expect it is estimating the parameters of the covariance function but this algorithm just calculates the likelihood.
> A: Thanks very much for pointing out this gap in Algorithm 1. The Algorithm 1 pseudocode really is missing the 7th step, which is to employ Newton’s method to solve the objective in equation 8 as computed by the pseudo code of the algorithm up to step 6 (we had been thinking this was implied, but upon second look it definitely was not clear enough).  We will rectify this in the revised draft.
>
> R:  Given the simple coupling structure with parameter tying, this problem seems to be closely related to estimating AR(N) style models (alpha and beta parameters) so I am surprised to see only a general treatment.
> In this problem, it seems much more natural to estimate theta and g recursively and concurrently, and there are very well known algorithms related to Kalman filtering. Please discuss.
> A: We agree in regards to employing algorithms such as Kalman filtering for potentially better estimates of the inverse covariance and covariance therein and thus potentially better gradient estimates. However, if done concurrently and recursively the gradient estimates of the first few images would be potentially erroneous because of a few number of samples being used to calculate the sample covariance being used in the optimization problem in equation 8. Hence, we initially choose to use 1000 queries in all from 20 images using 50 perturbations for each of them to get loss function values to estimate the MRF parameters so as to give the attack framework a warm start. At the same time,, we note that for generating the attacks when we get loss function values of perturbed images, those can be subsequently used to refine the estimation of MRF parameters. But, because of the nature of the optimization problem which requires using FFT techniques for an efficient computation framework, a recursive framework would anyways require to compute the objective function from scratch for any further loss function values. To sum it up, while employing recursive techniques could potentially better attack accuracies, it would be at the cost of expensive computation and an initial reasonable estimate at the beginning helps our framework to have a warm start.

---

> > ### Author Response · Authors · 2019-11-09
> > **Continued...**
> >
> > R:  The evaluation of the idea is not complete While it is certainly interesting to see that such a simple heuristic can achieve comparable performance on standard datasets over other black-box attack methods in the limited query budget regime, I would expect to see some experiments that illustrate the quality of the gradient estimator more closely (not only its final effectiveness for finding a search direction inside of an optimization method) and more strong justification of the proposed model.
> > A: Thanks for emphasizing this point, and after consideration we fully agree.  In the revised draft, we will add more experiments across different datasets to illustrate the performance of the gradient estimate and also provide empirical justification of employing a non-identity covariance structure for the gradients.   We are in the process of running these experiments now, and will post a further response and update when we include them.
> >
> > We would also like to point out that the effectiveness of our procedure is also partially dependent on the usage of FFT basis vectors as directional derivative directions for estimating the MAP estimate of the gradient which we illustrated in Figure 1. This is a significant departure from other black-box adversarial attacks and other classical zeroth order optimization schemes which typically rely on vectors from normal distribution or canonical basis vectors.
> >
> > R: As the entire contribution hinges on the observation that gradients across dimensions of an example as well as across examples in a dataset are correlated, it would have been also very informative to show estimates of autocorrelation functions of  the gradients on different datasets to justify the basic modelling choices.
> > A: This is a great suggestion.  We will add experiments to show estimates of autocorrelation functions across different datasets for justifying the basic modeling choices, and will post here when they are finished.
> >
> > After these edits and experiments, we hope that your main concerns with the experimentation, justification for the method, etc, are sufficiently addressed.  We will be following up soon with the additional results.

---

### Author Response · Authors · 2019-11-13
**Revision**

We would like to thank all the reviewers for the reviews and the insightful comments. We have uploaded a revised draft.

- The revised draft incorporates the gradient estimation performance on MNIST and ImageNet for the LeNet and VGG-16bn architectures in Section A.6
- We also added autocorrelation plots for the true gradients on MNIST and ImageNet for the LeNet and VGG-16bn architectures in Section A.7 for providing further justification of our gradient model.
- We have added the reference and discussion pertaining to it as recommended by Reviewer #1.
- We have added more discussion about our experimental results outlining the performance.

---

### Decision · Program_Chairs · 2019-12-19

**Decision:**

Reject

**Comment:**

This paper presents a Markov Random Fields (MRF) for generating adversarial examples in a black-box setting, where only it has access to loss function evaluations. The method exploits the structure of input data to model the covariance structure of the gradients. Empirically, the resulting method uses fewer queries than the current state of the art to achieve comparable performance. Overall, the paper has valuable contributions. The main issue is on empirical evaluation, which can be strengthened, e.g., by including results with multi-step methods and more thorough analysis of the estimated gradients.